# Revealing Interactions of Gut Microbiota and Metabolite in Confined Environments Using High-Throughput Sequencing and Metabolomic Analysis

**DOI:** 10.3390/nu16172998

**Published:** 2024-09-05

**Authors:** Ziying Wang, Haodan Xu, Xin Song, Zheng Chen, Guangqiang Wang, Yijin Yang, Beiwei Zhu, Lianzhong Ai, Chenxi Liu, Yaxuan Zhang, Yong Yang, Chuan Wang, Yongjun Xia

**Affiliations:** 1Naval Medical Center, Naval Medical University, Shanghai 200433, China; wziy0910@hotmail.com (Z.W.); xhdhit2016@163.com (H.X.); lcxlinda@163.com (C.L.); joycezhangyx@163.com (Y.Z.); 2Key Laboratory of Molecular Neurobiology of Ministry of Education, Shanghai 200433, China; 3School of Health Science and Engineering, Shanghai Engineering Research Center of Food Microbiology, University of Shanghai for Science and Technology, Shanghai 200093, China; daohongxuan@126.com (X.S.); 1015wanggq@163.com (G.W.); soliaran@163.com (Y.Y.); ailianzhong1@126.com (L.A.); wy931221@sina.com (Y.Y.); 4School of Food Science and Technology, Dalian Polytechnic University, Dalian 116034, China; sky_527@163.com (Z.C.); zhubeiwei@163.com (B.Z.)

**Keywords:** confined environment, gut microbiota, uric acid, beta-PHENYL-gamma-aminobutyric acid, multi-omics analysis

## Abstract

A confined environment is a special kind of extreme working environment, and prolonged exposure to it tends to increase psychological stress and trigger rhythmic disorders, emotional abnormalities and other phenomena, thus seriously affecting work efficiency. However, the mechanisms through which confined environments affect human health remain unclear. Therefore, this study simulates a strictly controlled confined environment and employs integrative multi-omics techniques to analyze the alterations in gut microbiota and metabolites of workers under such conditions. The aim is to identify metabolic biomarkers and elucidate the relationship between gut microbiota and metabolites. High-throughput sequencing results showed that a confined environment significantly affects gut microbial composition and clusters subjects’ gut microbiota into two enterotypes (Bla and Bi). Differences in abundance of genera *Bifidobacterium*, *Collinsella*, *Ruminococcus_gnavus*_group, *Faecalibacterium*, *Bacteroides*, *Prevotella* and *Succinivibronaceae* UCG-002 were significant. Untarget metabolomics analyses showed that the confined environment resulted in significant alterations in intestinal metabolites and increased the activity of the body’s amino acid metabolism and bile acid metabolism pathways. Among the metabolites that differed after confined environment living, four metabolites such as uric acid and beta-PHENYL-gamma-aminobutyric acid may be potential biomarkers. Further correlation analysis demonstrated a strong association between the composition of the subjects’ gut microbiota and these four biomarkers. This study provides valuable reference data for improving the health status of workers in confined environments and facilitates the subsequent proposal of targeted prevention and treatment strategies.

## 1. Introduction

A confined environment is a limited space with relative isolation from the outside world, strict restrictions on access and environmental particularity constructed for specific tasks. Typical scenarios include aerospace devices, submarines, martial tunnels, polar scientific research and home isolation. Staff who spend a long time in a confined environment are unable to perceive the changes of day and night, and it is difficult for them to vent their emotions reasonably, so they are prone to psychological disorders, triggering uncomfortable physiological symptoms such as disorders of the biological clock, insomnia, tightness in the chest and mental depression [1,2,3]. In addition, household appliances in confined environments, environmental microorganisms, temperature, humidity and air components can also have an impact on the health of staff members [4].

Research showed that persistent psychological stress is prone to causing personal psychological disorders, and affects the homeostasis of human gastrointestinal tract function and metabolic homeostasis through the bidirectional dialog of the gut microbiota–brain axis. In this process, the intestinal microbiota plays a very important regulatory role [5,6]. The balance of gut microbiota is not only the premise of maintaining normal intestinal function, but also the basis of nerve connection and biological signal molecule transmission between intestine and brain [7]. In addition, metabolites of gut microbiota, such as serotonin, dopamine, norepinephrine, short-chain fatty acids, γ-aminobutyric acid, histamine and acetylcholine, can act on the central nervous system (CNS) through nerve, endocrine and immune pathways, regulate the occurrence and development of nerves and the development of related central nervous diseases, and regulate the nervous system of the intestine itself. Therefore, gut microbiota can influence the function of both the brain and intestine through the gut microbiota–brain axis, leading to the occurrence of related diseases [8]. 

At present, many studies have proved the interaction between psychological stress and gut microbiota [9,10,11,12,13]. Barandouzi et al. [14] found that the intestinal microbial composition in depressive patients underwent significant changes compared to the healthy group, with a marked decrease in species diversity. Zhang et al. [15] discovered alterations in intestinal microbial diversity and metabolite profiles; several key differential gut microbial genera, including *Alistipes*, *Bacteroides* and *Parasutterella*, showed strong correlation with organic acids. Uric acid is an important indicator of body fluids. A study has shown that intestinal cells secrete uric acid to resist oxidative stress when stimulated by factors such as reactive oxygen species and hydrogen peroxide, and increase in uric acid is associated with gut microbiota imbalance [16].

Partial closed-environment–simulation experiments show that adjusting room light, temperature and air improves staff anxiety and increases subjects’ cognitive performance and work efficiency [17,18]. However, changes in human gut microbiota and their metabolites in confined environments, and the correlation between them, remain unclear. Therefore, investigating the intrinsic relationship between gut microbiota and metabolites in staff within confined environments is significant. In this paper, we analyzed the changes in intestinal microbial composition and metabolite profiles of staff members using high-throughput sequencing and untargeted metabolomics in a strict confined-space experimental environment, screened for metabolic markers, and elucidated the correlation between gut microbiota and metabolites. This research lays a theoretical basis for improving the health and work efficiency of staff in confined environments, and is conducive to proposing targeted strategies.

## 2. Materials and Methods

### 2.1. Subject

The study involved 45 healthy male participants aged 19–26 years old, with heights ranging from 163 to 186 cm and weights between 54 and 95 kg. The aim was to reduce potential confounding factors related to age-related physiological differences. All participants were free of cardiovascular diseases, hemorrhoids, infectious diseases, skin conditions and any other illnesses. They did not take alcohol, antibiotics or other drugs in the 7 days before the experiment and slept well to ensure that any effects observed were not affected by the original health problems. Each participant was fully informed of the experimental procedures and potential risks, and provided written informed consent. All experimental protocols were approved by the ethics committee of Naval Medical University under approval number 2023032302.

### 2.2. Confined Environment Simulation

The experiment was conducted in a simulated submarine cabin designed to replicate the conditions of a living room, featuring normal atmospheric pressure, room temperature and airtightness. The cabin has an effective volume of 200 m^3^, with temperature and relative humidity controlled to within ±0.5 °C and ±5% RH, respectively. Throughout the experiment, participants were prohibited from smoking and drinking, had a normal diet and a moderate workload every day. They had access to computers and fitness equipment, but there was no external network, and the cabin was kept in a state of information isolation. The confined environment experiment lasted 35 days. After leaving the simulated submarine cabin, participants were quarantined in a hotel for 16 days, during which conditions were like those at home, except for the restriction on leaving the premises.

### 2.3. Sample Collection

Fecal and urine samples were collected on the day before storehouse (A), the last day at the end of the confinement experiment (D) and 16 days after the hamper (E) and 15 parallel samples were set up for each treatment, labeled A1–15, D1–15, and E1–15, respectively. All samples were collected and placed on ice immediately, and subsequently stored at −80 °C for further analysis.

### 2.4. High-Throughput Sequencing and Bioinformatics Analysis

Fecal samples were prepared for gut microbiota diversity analysis. Total DNA from collected fecal samples was extracted using the PF Mag-Bind Stool DNA Kit (Omega Bio-tek, Norcross, GA, USA) according to the manufacturer’s protocols. The V3-V4 16S rRNA gene regions was amplified using the universal primers 338F (5′-ACTCCTACGGGGAGGCAGCA-3′) and 806R (5′-GGACTACHVGGGTWTCTAAT-3′). PCR amplification conditions were performed according to literature protocols [19]. The purified PCR products were subjected to quality quantification using Quantus™ Fluorometer (Promega, Madison, WI, USA) and then subjected to 16S rRNA gene sequencing on the Illumina MiSeq 2500 platform (Illumina, San Diego, CA, USA) at Majorbio Bio-Pharm Technology Co., Ltd. (Shanghai, China).

Gut microbiota analysis of the subjects was conducted with the Majorbio Online Cloud platform (https://cloud.majorbio.com/, accessed on 20 September 2022). Valid sequences with more than 97% similarity were clustered into one operational taxonomic unit (OTUs) using Uparse software (Version 7.0.1090). Representative sequences from each OTU were screened for further annotation. For representative bacterial OTU sequences, the SILVA database was used for annotation and classification. The α-diversity and β-diversity of each sample were assessed using Mothur software (Version 1.30.2) and QIIME software (Version 1.7.0), respectively, and the species diversity and abundance were compared using Kruskal–Wallis and Wilcox rank sum test. Linear regression analysis was performed using the Bray_Curtis distance algorithm based on genus level to evaluate the effects of differential metabolites on gut microbiota. Enterotype analysis of intestinal microbiota was conducted using the weighted_normalized_unifrac algorithm based on genus level. LEfSe was employed to perform linear discriminant analysis (LDA) on samples grouped by different gut types, identifying microbial communities or species significantly influencing gut type differentiation. The Spearman’s rank correlation coefficient was used to calculate the correlation between species relative abundance and differential metabolites, and the results were visualized using a heatmap.

### 2.5. Untargeted Metabolomics Analysis of Urine

All metabolites in urine were analyzed by untargeted metabolomics. In short, urine samples (100 μL) were mixed with 400 μL of extraction solution (acetonitrile:methanol = 1:1, *v*/*v*). After vortexing for 30 s, the samples were subjected to low-temperature ultrasonic extraction for 30 min (5 °C, 40 KHz). The mixture was centrifuged at 13,000× *g*, 4 °C for 10 min, and the supernatant was collected for subsequent LC-MS/MS analysis. Equal volumes of all sample metabolites were mixed to prepare a pooled quality control sample for assessing the reproducibility of the entire analysis and detection process. Metabolite detection of urine samples was performed using a SCIEX UPLC-Triple TOF 5600 (Thermo Fisher Scientific, MA, USA) system. Chromatographic conditions were as follows: UPLC column, ACQUITY HSS T3 column (100 mm × 2.1 mm i.d., 1.8 μm; Waters, Milford, MA, USA); column temperature, 40 °C; flow rate, 0.4 mL/min; injection volume, 10 μL; solvent system: solvent A consisted of 0.1% formic acid in water:acetonitrile (95:5, *v*/*v*) and solvent B consisted of 0.1% formic acid in acetonitrile:isopropanol:water (47.5:47.5:5, *v*/*v*/*v*); gradient elution. The samples were separated by HSS T3 column and then entered mass spectrometry (MS). MS equipped with an electrospray ionization (ESI) that can work in both positive and negative ion modes. ESI optimal parameters were set as follows: ionization source heating temperature 500 °C; ion spray voltage 5000 V (positive), −4000 V (negative).

The processing software Progenesis QI version 2.3 (Waters Corporation, Milford, CT, USA) was used to transform and analyze LC/MS raw data, including peak detection, comparison and identification, and finally a 3D data matrix in CSV format was obtained. After matching with metabolic public databases HMDB, Metlin and Majorbio databases, a data matrix was uploaded to Majorbio Online Cloud Platform for bioinformatics analysis. Significantly different metabolites were screened according to the variable importance in the projection (VIP) obtained by orthogonal least partial squares discriminant analysis (OPLS-DA) and the *p*-value generated by a paired-sample student’s t test. Differential metabolites were annotated for metabolic pathway analysis using the KEGG database to identify pathways involving these metabolites. Pathway-enrichment analysis was conducted using the Python software (version 3.12.5) package SciPy.Stats, and Fisher’s exact test was employed to identify the biological pathways most relevant to the experimental treatments.

## 3. Results

### 3.1. Enterotypes Analysis of Intestinal Microbiota in Subjects

The composition of intestinal microbiota in 60 subjects was detected via 16S rRNA high-throughput sequencing. A total of 2,972,632 sequences were obtained, and the average sequence length was 410 bp. These sequences were clustered into 822 operable taxa, including 13 phyla, 95 families and 258 genera. According to the floristic abundance at the genus level, the weighted_normalized_unifrac algorithm was used and PAM (Partitioning Around Medoids) clustering was carried out, and the best clustering K value was calculated with the Calinski–Harabasz (CH) index. The enterotype of gut microbiota was determined by principal coordinates analysis (PCoA) [20]. Enterotype analysis revealed that 45 participants could be classified into two enterotypes (Figure 1A). Type I was the *Blautia* type (Bla) and type II was the *Bifidobacterium* type (Bi). Although the two groups of participants exhibited a high degree of overlap in terms of intestinal microbiota types, there were significant differences in enterotype characteristics. Therefore, the subsequent analysis of gut microbiota is based on enterotype.

To further understand the microbial community diversity indicated by OTU, we calculated α-diversity to reflect species richness and diversity of individual samples. The α-diversity analysis of each enterotype group showed no significant differences in ACE index and Chao index between Bla group and Bi group (Figure 1B,E). Shannon and Simpson indexes are used to measure species diversity. Influenced by species richness and evenness within the sample community, higher values of the Shannon and Simpson indices indicate greater species diversity [21]. As shown in Figure 1C,D, the Shannon index of the Bla group was significantly higher than that of the Bi group, while the Simpson index of the Bi group was significantly higher than that of the Bla group, which may account for the differences in gut microbiota typing.

### 3.2. Effects of Confined Environment on Intestinal Microbiota of Different Enterotypes

To supplement the qualitative analysis, this work further examined the effects of confined environments on intestinal microbiota across different enterotypes at the phylum and genus levels, as shown in Figure 2A. The results indicated that the intestinal microbiota of the Bla enterotype group was dominated by Firmicutes, Bacteroides and Actinobacteriota, accounting for 95.87% of the total abundance, with Firmicutes and Bacteroides accounting for 74.14% and 14.81%, respectively. Different from the Bla enterotype group, Firmicutes, Actinobacteriota and Proteobacteria were the main phyla in the Bi enterotype group and accounted for 97.31% of the total, of which Firmicutes and Actinobacteriota accounted for 67.81% and 22.85%, respectively. At the genus level, the main components of the Bla enterotype group are *Blautia*, *Faecalibacterium*, *Bacteroides* and *Agathobacter*. And the main components of the Bi enterotype group are *Bifidobacterium*, *Blautia*, *Eubacterium*_*hallii*_group and *Escherichia*-*Shigella* (Figure 2B). Notably, the Bi enterotype group had higher levels of *Bifidobacterium* and *Blautia* compared to the Bla enterotype group. A bubble plot was used to further analyze the compositional structure of the gut microbiota, displaying the top 30 most abundant genera, where the size of the circles represented the differences in relative abundance. As shown in Figure 2C, most of the intestinal microbiota belonged to the Firmicutes phylum, and there were significant compositional differences between the Bla and Bi groups. The dominant bacteria also differed between the two groups, with the Bi group having higher levels of Actinobacteriota such as *Bifidobacterium* and *Collinsella*, compared to the Bla enterotype group.

As shown in Figure 2D, the upset plots analyzed the subject’s intestinal microbiota. The results showed that 134 core bacterial taxa were shared across the six groups. The number of Bi enterotype group was less than that of Bla group at genus level. There were no significant changes in microbial abundance between Bla and Bi enterotype groups before and after confined environment experiment. The source tracker pieplot indicated that 58.99% of the microbiota in the Bla_E group originated from the Bla_A group, 17.37% from the Bla_D group and only 2.74% of the microbiota was of unclear origin (Figure 2E). In the Bi_E group, approximately 50.81% of the microbiota originated from Bi_A, 31.33% from Bi_D and only 1.71% of the microbiota was of unclear origin (Figure 2E). These findings suggest that the microbiota of the two enterotype types before leaving the cabin are mainly from A and D groups, while a small number of microorganisms from unknown sources may be caused by airtight conditions. To sum up, a confined environment significantly altered the composition of the intestinal microbiota at both the phylum and genus levels, leading to notable changes in the dominant bacterial taxa in the subjects’ intestinal microbiota.

### 3.3. Differences in the Response of Different Intestinal Microbiota to Confined Environment

To further investigate the composition of intestinal microbiota in the two enterotypes within a confined environment, we evaluated the differences in microbiota at dominant phylum and genus levels, and identified species with significant intergroup differences. Hypothesis testing was performed on the species across different microbial community groups, as shown in Figure 3. There were highly significant changes (*p* < 0.01) in Fusobacteriota, Desulfobacterota, Bacteroidota and Actinobacteriota between the two enterotypes at phylum level (Figure 3A). The Bacteroidota phylum abundance in the Bla enterotype group was significantly higher than in the Bi group, whereas the Actinobacteriota phylum abundance in the Bi group was significantly higher than in the Bla group, which is consistent with the results shown in Figure 2. Genus-level analysis showed that the relative abundance of *Bifidobacterium*, *Eubacterium*_*hallii*_group, *Collinsella* and *Ruminococcus*_*gnavus*_group was significantly higher in Bi group than in Bla group (* *p* < 0.05, ** *p* < 0.01, *** *p* < 0.001) (Figure 3B). Meanwhile, *Faecalibacterium*, *Bacteroides*, *Fusicatenibacter*, *Agathobacter*, *Prevotella*, *Alistipes*, norank_f__*Eubacterium*_*coprostanoligenes*_group, *Succinivibronaceae* UCG-002, *Lachnospiraceae*_NK4A136_group, *Roseburia*, *Christensenellaceae*_R-7_group and *Parabacteroides* genera abundance was significantly lower in the Bi enterotypes group than in the Bla group (Figure 3B).

In addition, we used non-metric multidimensional scaling (NMDS) analysis to reduce the dimensionality of the intestinal microbiome data of subjects, and again verified the difference in intestinal microbiome composition between the Bi group and Bla group in a confined environment. The results showed that the intestinal microbiota composition of the Bi group changed throughout the experiment, while the Bla group showed almost no variation (Figure 3C). LDA results showed that the Bla_A group was significantly enriched in *Faecalibacterium*, *Alistipes* and *Lachnospiraceae*_NK4A136_group (Figure 3D). The Bla_D group, on the other hand, was notably enriched in the genera *Bacteroides*, *Succinivibronaceae* UCG-002 and *Roseburia*. In the Bla_E group, significant enrichment was observed in *Pyramidobacter*, *Anaerotruncus* and *Butyricimonas*. Conversely, the Bi_A group was significantly enriched in *Corynebacterium* and *Intestinibacter*. The Bi_D group showed significant enrichment in *Bifidobacterium* and *Streptococcus*, while the Bi_E group was significantly enriched in *Turicibacter*.

### 3.4. Untargeted Metabolomics Analysis

Untargeted metabolomics analysis was performed on urine samples before and after subjects entered and exited the cabin, and after data preprocessing, a total of 2637 metabolites were identified in this metabolome, of which 2311 and 700 metabolites were annotated to the HMDB and KEGG databases, respectively. The metabolic data were processed using OPLS-DA, as shown in Figure 4A; the metabolite compositions of groups A, D and E were significantly different from each other, and could be well distinguished from each other. KEGG pathway annotation was performed for all metabolites, and the top 20 ranked KEGG pathways are shown in Figure 4B. Biosynthesis of phenylpropanoids, tyrosine metabolism, tryptophan metabolism, ABC transporters, arginine and proline metabolism, bile secretion and steroid hormone biosynthesis are more active metabolic pathways. Figure 4C shows the categorization statistics of KEGG metabolic pathways, the metabolites annotated to the subjects were mainly concentrated in the categories of metabolism, human diseases and organismal systems, with the metabolism of amino acids (164), biosynthesis of other secondary metabolites (88), metabolism of cofactors and vitamins (52), lipid metabolism (164), biosynthesis of other secondary metabolites (88), metabolism of cofactors and vitamins (52), lipid metabolism (50) and digestive system (52). The results of HMDB compound categorization (class level) showed that compounds with relative abundance >2% were mainly in the categories of carboxylic acids and derivatives (17.80%), fatty acyls (13.02%), organooxygen compounds (11.66%) and prenol lipids (10.26%) (Figure 4D).

### 3.5. Differential Metabolite Analysis

The differential metabolite set Bla_Bi was established by grouping the two enterotypes according to the criteria (FC > 1, VIP > 1, *p* < 0.05), and a total of 198 differential metabolites were obtained. HDMB analysis of differential metabolites showed (class level) that compounds with relative abundance > 2% were mainly found in carboxylic acids and derivatives (13.33%), organooxygen compounds (13.33%), fatty acyls (11.11%) and prenol lipids (10.00%) categories (Figure 5A), which differed somewhat from the overall metabolite categorization (Figure 4D). OPLS-DA analysis showed that the differential metabolites of subjects in the Bla group in the confined environment were well differentiated from those of the Bi group, suggesting that confined-space living had a significant effect on subject metabolism (Figure 5B). The analysis results of the volcano plot indicated that a total of 198 metabolites were significantly different (*p* < 0.05, FC > 1) in the Bla_ vs. Bi_ group, with 104 metabolites upregulated and 94 metabolites downregulated in the Bla group compared to the Bi group (Figure 5C). The KEGG database was used to annotate the differential metabolites, and the results indicated that most of the differential metabolites were annotated as amino acid metabolic pathways (Figure 5D). Further KEGG enrichment analysis of differential metabolites revealed that metabolic pathways such as cholesterol metabolism, secondary bile acid biosynthesis, primary bile acid biosynthesis, bile secretion, tyrosine metabolism, phenylalanine metabolism, α-linolenic acid metabolism and acridone alkaloid biosynthesis were significantly enriched (*p* < 0.05) (Figure 5E). In the enriched pathway, the differential metabolites were mainly concentrated in the piece of amino acid metabolism and bile acid-related metabolism. As shown in Figure 5F, the heat map analysis of the top 30 differential metabolites of VIP rankings showed that the Bla group metabolites were significantly different from the Bi group. The 25 of compounds such as mesobilirubinogen, URSINIC ACID, glycocholic acid, glycocholic acid, and virginiamycin m1 were significantly upregulated in the Bla group (Figure 5F). While in Bi group, prostaglandin D1, 5,5-Diisopropyl-2,2′-dimethylbiphenyl-3,3′,4,4′-tetrone, enterolactone 3′-sulfate, 4-hydroxytriazolam, and cholic acid compounds were significantly upregulated (Figure 5F). 

### 3.6. Effect of Confined Environment on the Relative Abundance of Uric Acid

In this work, we analyzed the relative abundance changes of uric acid before and after the subjects were subjected to the confined environment experiment, and the result was shown in Figure 6. In particular, the relative uric acid content of the Bi group was significantly higher than in Bla group throughout the experiment (Figure 6A). After living in the confined environment, the relative uric acid content in both the Bla and the Bi group subjects showed a significant increase, and sometime after leaving cabin, the relative uric acid content decreased significantly and significantly compared with that in the cabin, and recovered to the level before the cabin. The above results suggest that the confined environment may lead to an imbalance of uric acid content in humans (Figure 6B,C).

### 3.7. Correlation Analysis of Intestinal Microbiota with Metabolites

The differential metabolites were ranked by VIP value, the top 25 differential metabolites were taken to be analyzed and screened for potential biomarkers, and the results are shown in Figure 7. The metabolites with TOP25 VIP values were analyzed by receiver operating characteristic (ROC) curve analysis, and five metabolites with ROC > 0.80 were obtained, which had VIP values ranging from 3.39 to 4.93. The HMDB classification based on the class level was mainly prenol lipids, benzene and substituted derivatives, and carboxylic acids and derivatives (Figure 7A). Figure 7B showed a comparison of the differences of these five metabolites in the two enterotypes, and all five differential metabolites were significantly upregulated compared to the Bi group. The KEGG pathway-annotation results of the differential metabolites in Figure 5 have shown that amino acid metabolism and lipid metabolism are more active during metabolism, and in combination with the ROC analysis in Figure 7C, it was hypothesized that the amino acid and lipid metabolites in the metabolites (beta-PHENYL-gamma-aminobutyric acid, (3S,5R,6R,6′S)-6,7-didehydro-5,6-dihydro-3,5,6′-trihydroxy-13,14,20-trinor-3′-oxo-beta,epsilon-caroten-19′,11′-olide 3-acetate, and lyciumoside VIII) as well as uric acid may be potential biomarkers.

Figure 8 shows the linear regression analysis of intestinal microbiota in different groups of participants with uric acid, beta-PHENYL-gamma-aminobutyric acid, (3S,5R,6R,6′S)-6,7-didehydro-5,6-dihydro-3,5,6′-trihydroxy-13,14,20-trinor-3′-oxo-beta, epsilon-caroten-19′,11′-olide 3-acetate and lyciumoside VIII metabolites were analyzed by linear regression. Significant correlations were found between the Shannon index of gut microbiota and four potential biomarkers in both groups of gut types (Figure 8A–D). Among them, uric acid showed a significant negative correlation, while the remaining three metabolites showed a significant positive correlation (*p* < 0.05).

Spearman’s correlation coefficient was used to analyze the relationship between intestinal microbiota and uric acid, beta-PHENYL-gamma-aminobutyric acid, (3S,5R,6R,6′S)-6,7-didehydro-5,6-dihydro-3,5,6′-trihydroxy-13,14,20-trinor-3′-oxo-beta, epsilon-caroten-19′,11′-olide 3-acetate and lyciumoside VIII were correlated among the four metabolites (Figure 9A). Most genera abundance correlated significantly with the biomarkers. Among them, the genera *Bifidobacterium*, *Streptococcus*, *Lactobacillus*, *Fusicatenibacter*, *Ruminococcus*_*gnavus*_group, *Monoglobus* and *Enterobacter* were positively correlated with the uric acid level. The genera *Bacteroides*, *Faecalibacterium*, *Dorea*, *Escherichia-Shigella*, *Succinivibronaceae* UCG-002, *Coprococcus* and norank_f__*Eubacterium_coprostanoligenes*_group were negatively correlated with uric acid level. The remaining three markers showed a consistent trend of species correlation, showing a negative correlation with *Bifidobacterium*, *Ruminococcus_gnavus*_group and a positive correlation with the relative abundance of the genera *Faecalibacterium*, *Bacteroides*, *Prevotella* and *Succinivibronaceae* UCG-002. The above results suggest that these genera may significantly influence the structure of intestinal microbiota.

Multivariate association with linear models (MaASLin) analysis is an approach that effectively identifies multivariate associations between clinical data and microbiomic profiles [22]. In this study, MaAsLin analysis confirmed that the genera *Bifidobacterium*, *Succinivibronaceae* UCG-002 and *Enterococcus* were significantly correlated with potential biomarkers that may influence the health status of the subjects (Figure 9B–E).

## 4. Discussion

Confined environments are isolated and closed extreme working environments, where living for a long period of time can lead to a series of problems, thus greatly affecting work efficiency and physical and mental health [23,24]. Confined environments, often found in aerospace, deep-sea and polar research situations, are integrated living and working environments. In such situations, the occupants are subjected to workloads, noise, work rhythms and other factors, leading to metabolic disorders. This in turn acts on the bidirectional dialog of the gut microbiota–brain axis, inducing a range of disorders [25,26]. Current research on the health impacts of confined environments predominantly focuses on ergonomics, behavior and other related factors, while studies on the composition of gut microbiota are relatively limited. In a recent review, the authors explored the impact of various work environments on the human gut microbiota [27]. However, the gut microbiota is closely linked to gastrointestinal metabolic homeostasis. To address this, we explored the changes in the intestinal microbiota and metabolites of staff in a confined environment using a multi-omics coupling technique and statistically analyzed the correlations in this study.

Enterotype classification is the clustering of different samples with similar structures of dominant microbiota into one group by means of statistical clustering, which provides a reliable tool for understanding the microbial diversity in healthy individuals and patients [28]. Microbial diversity analysis revealed significant differences in the intestinal microbiota of the subjects, which could be categorized into Bi and Bla enterotype groups. This result is inconsistent with traditional enterotype analysis methods but aligns with findings from a recent study [18,29]. In addition, it was found that a confined environment significantly altered community abundance and microbiota composition in both groups of enterotype. The study suggests that differences in the intestinal microbiota may influence metabolic phenotypes as well as responses to diet, stress and surrounding environment, thereby influencing physical and mental health [30,31,32]. The significant enrichment of the genera *Bifidobacterium*, *Eubacterium_hallii*_group, *Collinsella* and *Ruminococcus_gnavus*_group in the Bi group, and the lower relative abundance of *Succinivibronaceae* UCG-002 may have made the subjects in this group more adapted to the closed environment compared to the Bla enterotype group [19]. Metabolomic analyses showed that the confined environment significantly altered the types and levels of metabolites in both groups of enterotype samples. In terms of metabolic pathways, the differences were mainly focused on amino acid metabolism and bile acid-related metabolism. In addition, we hypothesized that uric acid, beta-PHENYL-gamma-aminobutyric acid, (3S,5R,6R,6′S)-6,7-didehydro-5,6-dihydro-3,5,6′-trihydroxy-13,14,20-trinor-3′-oxo-beta, epsilon-caroten-19′,11′-olide 3-acetate and lyciumoside VIII 4 differential metabolites as potential biomarkers.

Uric acid, also known as trioxopurine, is the final product of purine metabolism [33]. A study indicated that intestinal cells secrete large amounts of uric acid to resist external environmental stimuli, and increased uric acid level is associated with an imbalance in intestinal microbiota [16]. In this work, we revealed that the confined environment had different effects on uric acid levels in participants in the Bla and Bi enterotype groups. Confined environments can significantly increase uric acid levels, possibly because of metabolic abnormalities, which is consistent with results of Chen et al. [19]. Additionally, subjects’ uric acid levels were negatively correlated with the abundance of bacterial genera such as *Succinivibronaceae* UCG-002, *Coprococcus*, *Ruminococcus* and *Fusicatenibacter*, which were reported in the study to possibly affect uric acid metabolism [34]. Furthermore, we found that *Bifidobacterium* abundance was significantly higher in the Bi group compared to the Bla group and it exhibited a strong positive correlation with uric acid level. It has been shown that *Bifidobacterium* can promote the degradation of uric acid in the intestine [35,36]. Therefore, we can hypothesize that the elevated uric acid levels promote an increase of *Bifidobacterium* abundance to maintain the balance of uric acid content in the body.

Beta-PHENYL-gamma-aminobutyric acid, also known as phenibut, is an agonist used clinically to treat anxiety and alcohol withdrawal symptoms. In Europe and America, it is marketed as a nutritional supplement for improving sleep [37]. Prolonged social isolation in closed environments may lead to emotional abnormalities, poor interpersonal skills, insomnia, decreased appetite, etc. [38]. In our results, the genera *Bacteroides*, *Prevotella*, *Faecalibacterium*, *Ruminococcus*, *Fusicatenibacter* and *Succinivibronaceae* UCG-002 were positively correlated with beta-PHENYL-gamma-aminobutyric acid level. A study has reported that *Succinivibronaceae* UCG-002, *Ruminococcus* and CAG-352 are closely related to insomnia [39]. *Succinivibronaceae* UCG-002 in the gut is the main bacterial genus positively associated with mediating chronic insomnia and coronary microvascular dysfunction [40] and is strongly associated with decreased appetite [10]. And the genus *Fusicatenibacter* may be associated with memory decline or cognitive impairment [41]. No other two metabolites have been reported on confined environments, so their use as potential biomarkers awaits further study.

Study on the effects of closed environments on human health is still in its infancy in our country [19]. This type of test is usually limited by several experimental conditions, such as small number of subjects, short experimental period and the physical condition of the subjects in the confined environment. Although the number of subjects in this study was only 45, this paper provides a database for improving the health and work efficiency of workers in confined space environments. Further in-depth analysis of gut types will require larger sample sizes, particularly focusing on functional analysis of core intestinal microbiota. It is anticipated that regulating specific microbial communities may help enhance human health in confined environments.

## 5. Conclusions

In this work, we investigated the effects of a confined space on gut microbiota and metabolites. Microbial diversity and metabolomic analyses were utilized to document the response of subjects with different enterotypes to a confined environment. Significant differences in species composition abundance and metabolite levels were observed between Bi and Bla enterotype-type subjects after experiencing a confined environment. Uric acid, a potential biomarker, showed a significant increase in the levels of subjects while living in confined spaces and was significantly correlated with the genera *Bifidobacterium*, *Succinivibronaceae* UCG-002, *Coprococcus*, *Ruminococcus* and *Fusicatenibacter*. In addition, beta-PHENYL-gamma-aminobutyric acid showed significant correlation with the genera *Bacteroides*, *Prevotella*, *Faecalibacterium*, *Ruminococcus*, *Fusicatenibacter*, and *Succinivibronaceae* UCG-002 as potential biomarkers. The metabolites (3S,5R,6R,6′S)-6,7-didehydro-5,6-dihydro-3,5,6′-trihydroxy-13,14,20-trinor-3′-oxo-beta,epsilon-caroten-19′,11′-olide 3-acetate and lyciumoside VIII as potential biometabolites remain to be further investigated.

## Figures and Tables

**Figure 1 nutrients-16-02998-f001:**
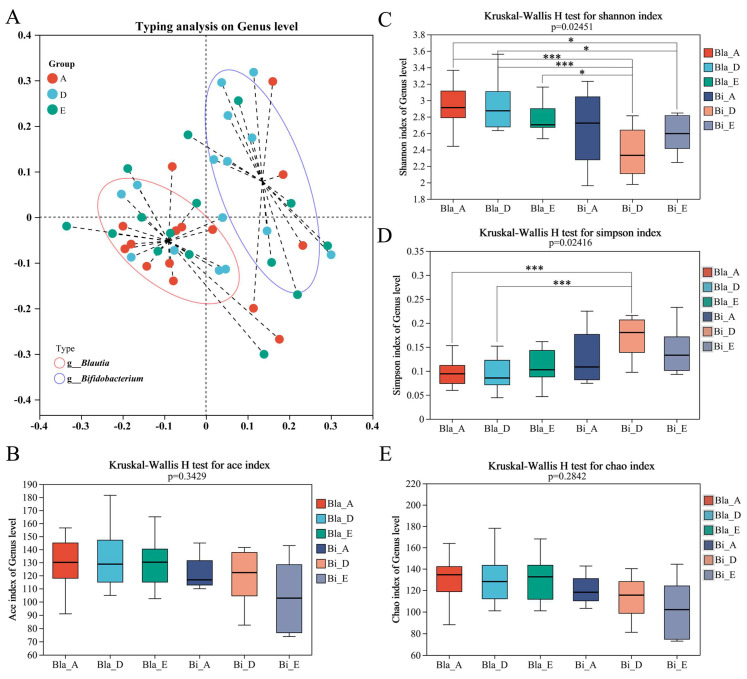
Analysis of enterotype clustering and α-diversity changes in closed environment. (**A**) Enterotype analysis of gut microbiota (A: before entering the cabin; D: in the cabin; E: out of the cabin). (**B**–**E**) Box plots plotted with the ACE, Shannon, Simpson and Chao indices, respectively. Kruskal-Wallis rank sum test was used to compare the group differences, * *p* < 0.05, *** *p* < 0.001.

**Figure 2 nutrients-16-02998-f002:**
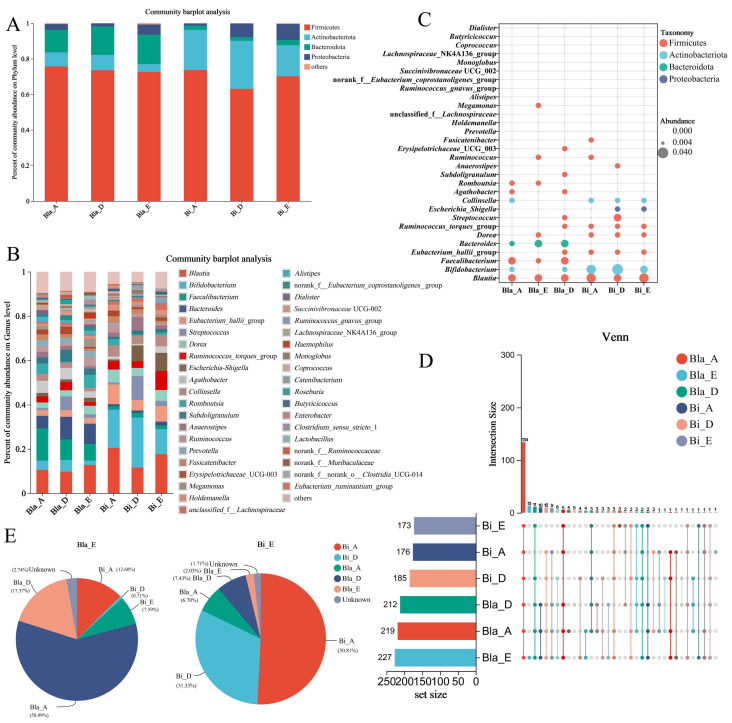
Alterations in intestinal microbial composition in closed environments. (**A**,**B**) Relative abundance barplot of intestinal microbiota based on phylum and genus levels; (**C**) bubble plot analysis of intestinal microbiota in different enterotype groups; (**D**) upset plots analysis of intestinal microbiota in different enterotype groups; (**E**) source tracker pieplot of intestinal microbiota in different enterotype groups.

**Figure 3 nutrients-16-02998-f003:**
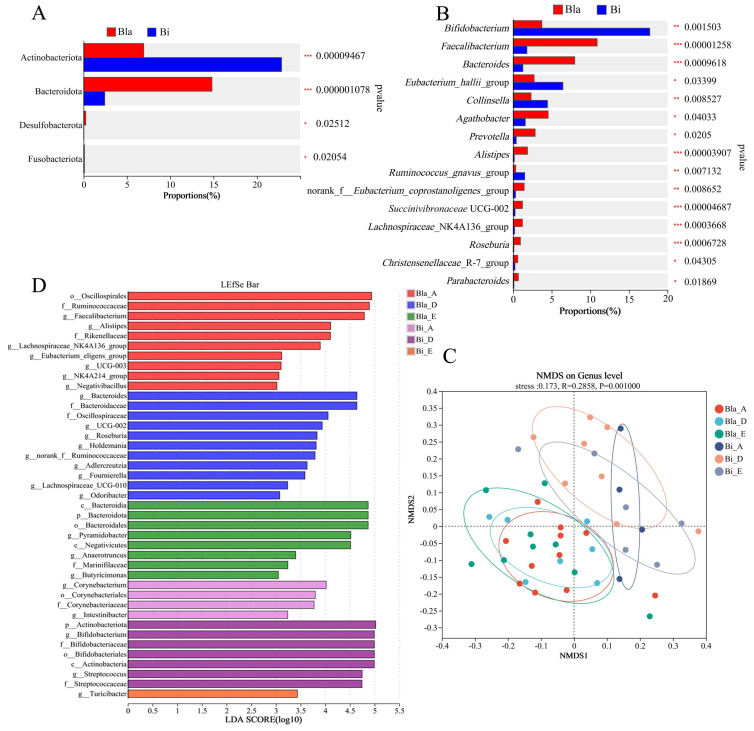
Significant differences analysis of intestinal microbiota in closed environments. (**A**,**B**) Analysis of interspecific difference analysis based on phylum (**A**) and genus (**B**) level; (**C**) NMDS analysis of gut microbiota; (**D**) significantly different species LDA score results (threshold value 2.5). Wilcoxon rank sum test was used to compare the group differences, * *p* < 0.05, ** *p* < 0.01, *** *p* < 0.001.

**Figure 4 nutrients-16-02998-f004:**
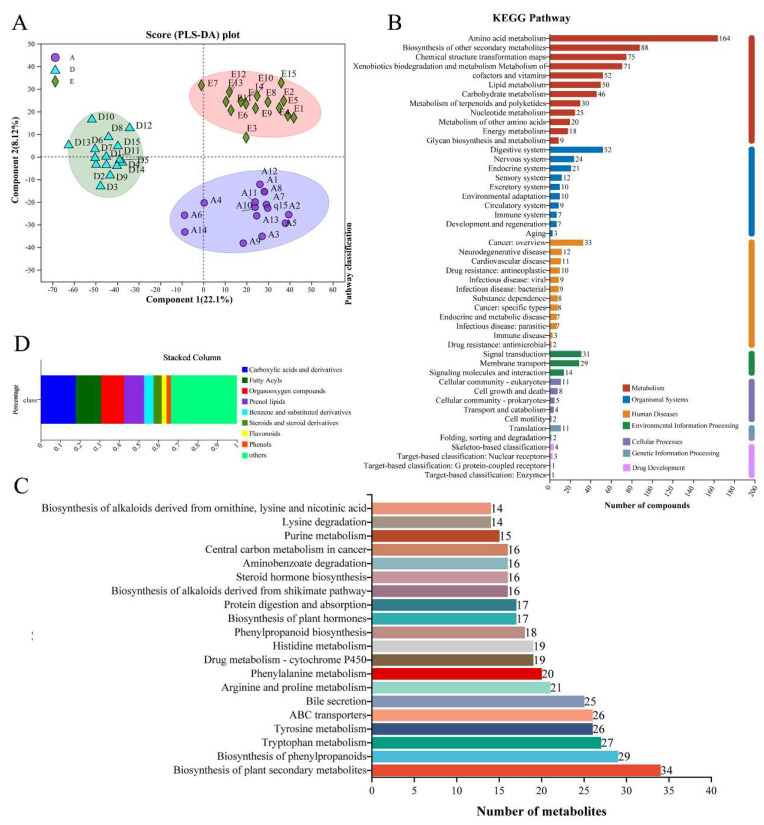
Annotations on metabolites in confined environment. (**A**) Comparative analysis of subjects’ samples; (**B**) metabolite annotation pathway statistics; (**C**) statistics of important pathways and metabolites in the top 20; (**D**) annotations on metabolites in HWDB database.

**Figure 5 nutrients-16-02998-f005:**
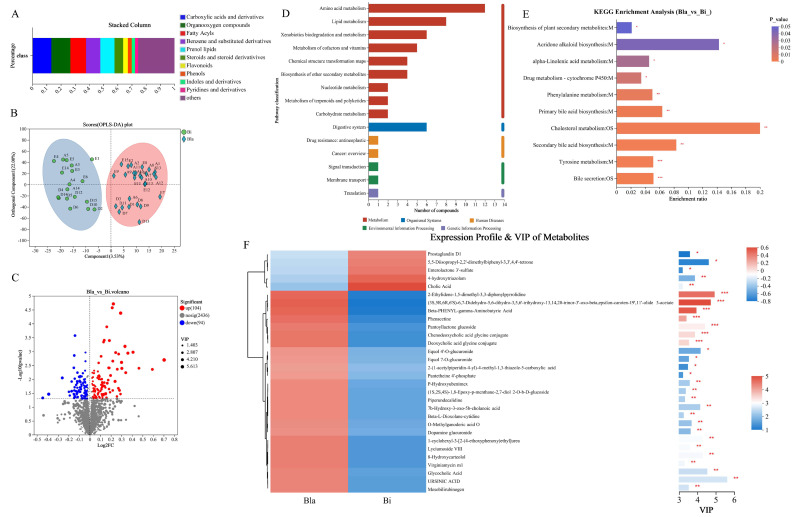
Analysis of significant difference of metabolites based on enterotypes. (**A**) Annotations on differential metabolites in HWDB database; (**B**) comparative analysis of different enterotype samples; (**C**) volcano map of different metabolites in two enterotype samples; (**D**) KEGG annotation analysis of differential metabolites; (**E**) KEGG enrichment analysis of differential metabolites; (**F**) analysis of VIP value of differential metabolites. * *p* < 0.05, ** *p* < 0.01, *** *p* < 0.001.

**Figure 6 nutrients-16-02998-f006:**
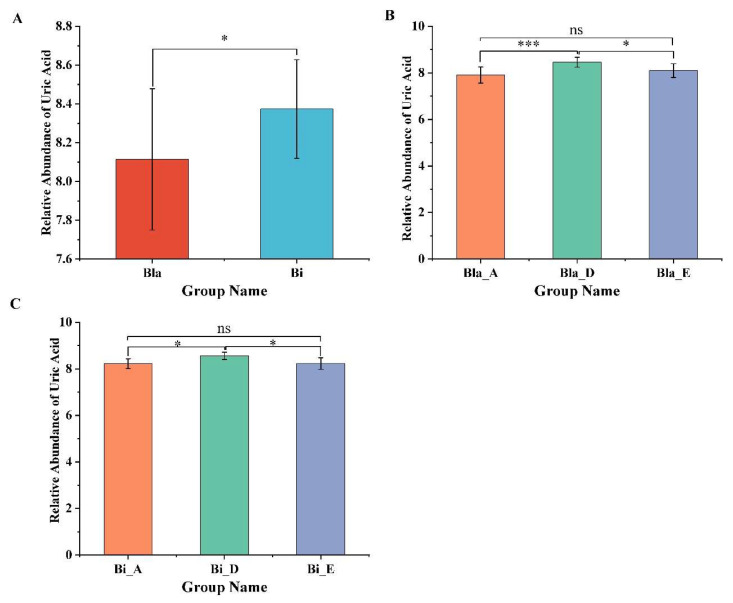
Changes in relative abundance of uric acid in confined environment. (**A**) changes in relative abundance of uric acid in different enterotypes; (**B**,**C**) changes in relative abundance of uric acid in Bla and Bi enterotype at different stages, respectively. LSD multiple test was used to compare the group differences, * *p* < 0.05, *** *p* < 0.001, ns represents no significance.

**Figure 7 nutrients-16-02998-f007:**
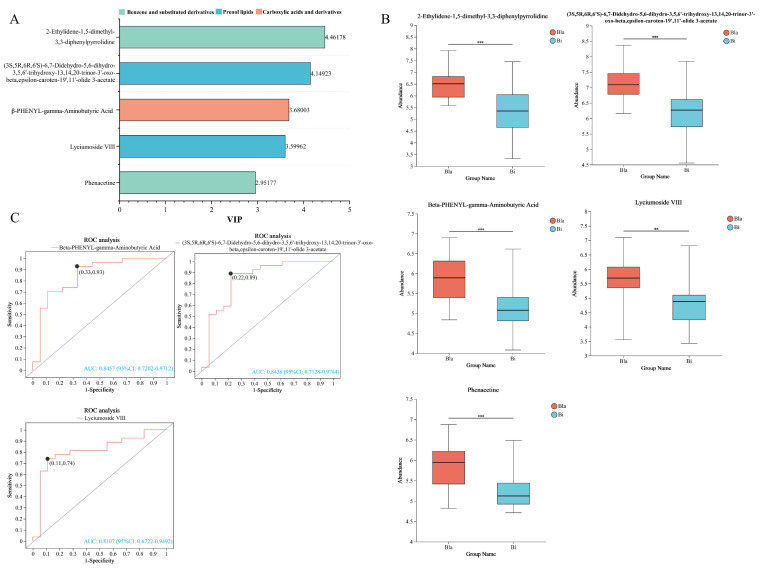
Screening of potential biomarkers. (**A**) HWDB annotations on five metabolites; (**B**) the difference of relative abundance of five metabolites between two enterotypes; (**C**) ROC analysis. Unpaired student’s t-test was used to compare the group differences, ** *p* < 0.01, *** *p* < 0.001.

**Figure 8 nutrients-16-02998-f008:**
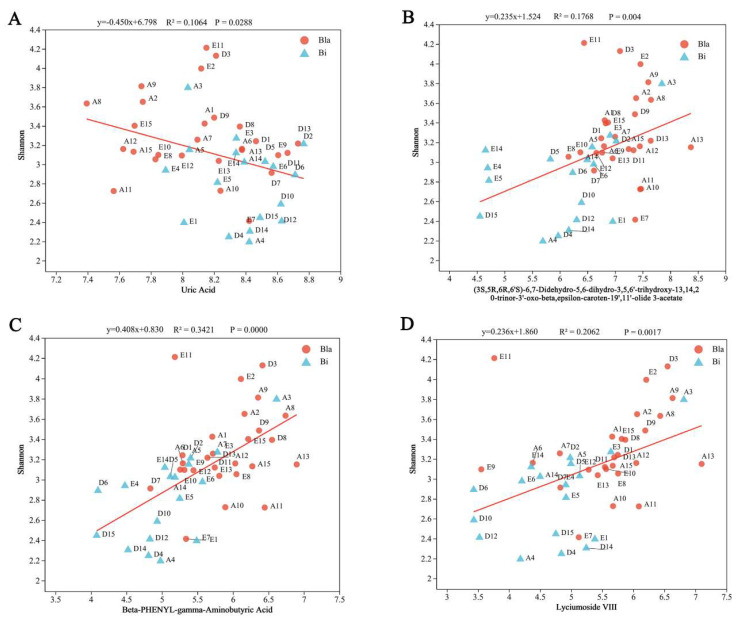
Linear regression analysis of four different biomarkers and gut microbiota characteristics. (**A**) Linear regression of gut microbiota with uric acid. (**B**) Linear regression of gut microbiota with (3S,5R,6R,6′S)-6,7-didehydro-5,6-dihydro-3,5,6′-trihydroxy-13,14,20-trinor-3′-oxo-beta, epsilon-caroten-19′,11′-olide 3-acetate. (**C**) Linear regression of gut microbiota with beta-PHENYL-gamma-aminobutyric acid. (**D**) Linear regression of gut microbiota with lyciumoside VIII.

**Figure 9 nutrients-16-02998-f009:**
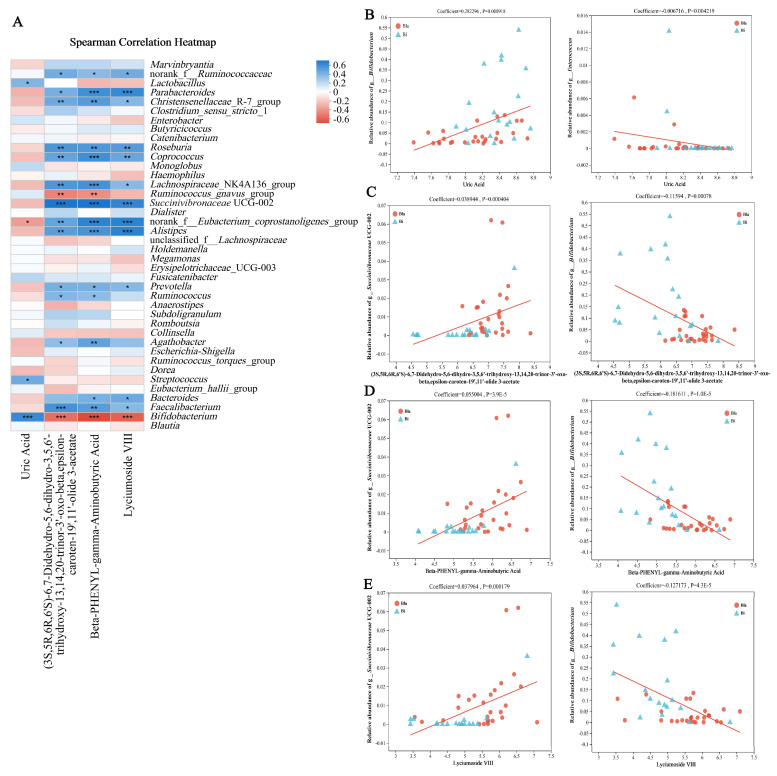
Correlation analysis between four different biomarkers and the composition of intestinal microbiota. (**A**) Correlation analysis between intestinal microbiota and biomarkers; (**B**–**E**) MaAslin analysis between intestinal microbiota and biomarkers. The asterisk in the heat map represents the significant *p* value, * *p* < 0.05, ** *p* < 0.01, *** *p* < 0.001.

## Data Availability

The data presented in this study are available on request from the corresponding author. The data are not publicly available due to secrecy agreement.

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
