# Peer review of "Revealing Interactions of Gut Microbiota and Metabolite in Confined Environments Using High-Throughput Sequencing and Metabolomic Analysis"

_nutrients, 2024, doi:10.3390/nu16172998_

Round 1

Reviewer 1 Report

Comments and Suggestions for Authors

The study conducted by Wang et al. evaluated the effect of confinement on changes in gut microbiota and metabolites using a combined multiomics technique to analyze metabolic markers and associate them with gut microbiota.

It is important that the study follows the consortium guidelines. Describe the eligibility criteria in full. Was the study blinded?

Was a sample size calculation performed beforehand?

How was randomization into groups performed? Describe the control group (non-confined);

It is important to better describe the sample characterization, weight, age, medications in use, associated diseases, i.e., factors that may alter the homogeneity of the data;

Although the study used analyses with modern technologies, such as multiomics, it is essential that its conduct be in accordance with some guideline for clinical studies, thus reducing bias.

Author Response

The study conducted by Wang et al. evaluated the effect of confinement on changes in gut microbiota and metabolites using a combined multiomics technique to analyze metabolic markers and associate them with gut microbiota.

  • We thank you for your finding interesting in our manuscript and for the accurate summary of the results of the paper.
  1. It is important that the study follows the consortium guidelines. Describe the eligibility criteria in full. Was the study blinded?

A: Thank you for your insightful comments. We understand the importance of adhering to consortium guidelines and have ensured that our study complies with them. In response to your request, we have now provided a detailed description of the eligibility criteria in Subject section of the revised manuscript. In this study, all experiments were conducted in accordance with ethics committee approval, approval number 2023032302. Regarding the blinding process. Before participating in the experiment, the subjects have understood the process and risks of this experiment, and signed the informed consent form. We believe these revisions will address your concerns and improve the clarity and rigor of our study.

  1. Was a sample size calculation performed before hand?

A: Thank you for your valuable feedback. A sample size calculation was not performed prior to this study due to special information on subjects' identity, confidentiality of experimental sites. However, we believe that the sample size used in this study is sufficient to provide meaningful insights and serves as a foundation for further research. We acknowledge the importance of conducting a sample size calculation and will incorporate it in future studies to enhance the statistical power and robustness of the findings.

  1. How was randomization into groups performed? Describe the control group (non-confined);

A: We sincerely thank the reviewer for careful reading. The subjects of this study were 45 healthy men. The experiment was conducted in submarine environment simulation cabin. The subjects could use computers, fitness equipment, etc., but there was no external network, and the submarine environment simulation cabin was in a state of information isolation to ensure that all the environment of the subjects was consistent. This study did not set up a non-closed group, and the various health indicators of the subjects were analyzed by comparing each other before, during and after entering the warehouse. This can better illustrate the impact of a confined environment on human health.

  1. It is important to better describe the sample characterization, weight, age, medications in use, associated diseases, i.e., factors that may alter the homogeneity of the data;

A: We sincerely thank you for your feedback. We describe the sample characteristics in detail in Subject section, including age, height, weight, medications used and associated diseases, etc., in Line 89-96.

  1. Although the study used analyses with modern technologies, such as multiomics, it is essential that its conduct be in accordance with some guideline for clinical studies, thus reducing bias.

A: Thank you for your valuable feedback. We agree that following established guidelines for clinical studies is crucial in reducing bias and ensuring the reliability of our results. In response to your suggestion, we have carefully reviewed our study's methodology and ensured that it aligns with recognized guidelines, such as insert specific guidelines, e.g., CONSORT, STROBE, etc. We appreciate your insight and believe these adjustments enhance the rigor and transparency of our work.

Reviewer 2 Report

Comments and Suggestions for Authors

This paper "Revealing Interactions of Gut Microbiota and Metabolite in Confined Environments Using High-throughput Sequencing and Metabolomic Analysis" outlines an important and timely study with potentially significant implications for health in confined environments. However, it could be improved by providing more context, clarifying the scientific methods, and emphasizing the novelty and applicability of the findings. Refining the language and including more detailed explanations of technical terms would also make the paper more accessible and compelling. Several parts are not clear:

-The authors should improve the introduction by better describing the role of the microbial in human health and especially the brain-microbiota axis.

-Describe better the rationale with which the study subjects were selected.

-In the Conclusions, we will better define the role that the discovery of this difference in microbial population could have and possible clinical scenarios.

-Furthermore, it is not clear how the confined environment negatively influences the organism from the evidence produced by the experiments: it is necessary to better articulate this concept through a dedicated paragraph.

-Also Percent match: 39% is too high, definitely decrease the percentage.

Comments on the Quality of English Language

Moderate editing of English language required.

Author Response

This paper "Revealing Interactions of Gut Microbiota and Metabolite in Confined Environments Using High-throughput Sequencing and Metabolomic Analysis" outlines an important and timely study with potentially significant implications for health in confined environments. However, it could be improved by providing more context, clarifying the scientific methods, and emphasizing the novelty and applicability of the findings. Refining the language and including more detailed explanations of technical terms would also make the paper more accessible and compelling. Several parts are not clear:

  • Thank you very much for your advice. We have revised the manuscript according to your professional advice. Would you please check them?
  1. The authors should improve the introduction by better describing the role of the microbial in human health and especially the brain-microbiota axis.

A: Dear reviewer, thank you for your careful review and constructive suggestions regarding our manuscript. We have revised the manuscript in accordance with the comments, in Line 50-63.

  1. Describe better the rationale with which the study subjects were selected.

A: Thanks for your comments concerning our paper. Long-term exposure to confined environments can lead to a state of chronic stress, which can easily induce psychological disorders and have a negative impact on staff health, living conditions and work efficiency. However, changes in gut microbiota in confined environments and how they correlate with health indicators have remained unclear. Therefore, we analyzed changes in gut flora composition and metabolites in subjects before, during, and after isolation, laying the groundwork for studying the effects of confined environments on human health.

  1. In the Conclusions, we will better define the role that the discovery of this difference in microbial population could have and possible clinical scenarios.

A: We feel great thanks for your professional review work on our article.

  1. Furthermore, it is not clear how the confined environment negatively influences the organism from the evidence produced by the experiments: it is necessary to better articulate this concept through a dedicated paragraph.

A: We really appreciate your rigorous work attitude. Thanks for your valuable comments again. Most current research suggests that the negative effects of confined environments on organisms are primarily the induction of psychological disorders, which are often characterized clinically by dysbiosis of the intestinal flora and abnormal changes in metabolites [1-6]. In this study, we conducted a joint analysis of gut microbiota and metabolites to parse out the effects of confined environment on the subjects. Some of these subjects may experience anxiety, memory loss, and anorexia, etc., in which case his gut flora composition and some metabolite levels will definitely show significant differences compared to other subjects, as explained in the Discussion in the context of the literature.

  1. Also Percent match: 39% is too high, definitely decrease the percentage.

A: Thank you for kindly reminding us. We have revised the manuscript.

  1. Moderate editing of English language required.

A: Thanks for your suggestion. We have tried our best and invited professionals to polish the grammar of the entire manuscript. We hope the revised manuscript will be acceptable to you.

References

[1] Chen, Z.; Wang, Z.; Li, D.; Zhu, B.; Xia, Y.; Wang, G.; Ai, L.; Zhang, C.; Wang, C. The Gut Microbiota as a Target to Improve Health Conditions in a Confined Environment. Front. Microbiol. 2022, 13, 1067756.

[2] Hao, Z., Meng, C., Li, L. et al. Positive Mood-related Gut Microbiota in a Long-Term Closed Environment: a Multiomics Study based on the “Lunar Palace 365” Experiment. Microbiome 2023, 11, 88.

[3] Song, X.; Wang, Z.; Xia, Y.; Chen, Z.; Wang, G.; Yang, Y.; Zhu, B.; Ai, L.; Xu, H.; Wang, C. A Cross Talking between the Gut Microbiota and Metabolites of Participants in a Confined Environment. Nutrients 2024, 16, 1761.

[4] Hao, Z.K.; Zhu, Y.Z.; Feng, S.Y.; et al. Effects of Long-Term Isolation on the Emotion Change of “Lunar Palace 365” Crewmembers. Sci. Bull. 2019, 64, 881–4.

[5] Hao, Z.K.; Feng, S.Y.; Zhu, Y.Z.; et al. Physiological Phenotypes and Urinary Metabolites associated with the Psychological Changes of Healthy Human: A Study in “Lunar Palace 365.” Acta Astronaut. 2020, 176, 13–23.

[6] Turroni, S.; Rampelli, S.; Biagi, E.; et al. Temporal Dynamics of the Gut Microbiota in People Sharing a Confined Environment, a 520-day Ground-Based Space Simulation, MARS500. Microbiome 2017, 5, 39.

Reviewer 3 Report

Comments and Suggestions for Authors

The paper presented to me for review is interesting from a medical point of view. The research was planned correctly. Advanced biochemical methods were used in the study.

The statistical methods used do not raise my objections.

The authors presented the results of their research in the form of 9 figures 

On the basis of their research, they drew the right conclusions

In their work, the authors cited 35 current scientific papers on the subject of the study.

In my opinion, the work is very good and interesting so I propose to accept for publication in the current version

Author Response

The paper presented to me for review is interesting from a medical point of view. The research was planned correctly. Advanced biochemical methods were used in the study.

The statistical methods used do not raise my objections.

The authors presented the results of their research in the form of 9 figures.

On the basis of their research, they drew the right conclusions.

In their work, the authors cited 35 current scientific papers on the subject of the study.

In my opinion, the work is very good and interesting so I propose to accept for publication in the current version.

A: Dear reviewer, we appreciate your summary of the manuscript and encouraging comment.

Reviewer 4 Report

Comments and Suggestions for Authors

This is a research article that focuses on the study of confined environments and their impact on human health , especially gut microbiota and its metabolites by advanced combined multi-omic technics. The abundance of Bifidobacterium, Collinsella, Faecalibacterium, and Bacteroides and identification of specific enterotypes in specific confined environments shows how confined environments impact the microbiota composition. Yet, metabolomic analysis showes increase in aminoacids and bile acids pathways .The authors seek to find potential biomarkers for developing targeted strategies to mitigate negative disease effects and protect people working in such environments. 

It is a well written paper with all recent bibliography in the field. The experimental part  is well designed . Their data were well presented in tables and images and extensively discussed. The researcher's team seems to be qualified and experienced in this scientific area .

The paper could be published in its present as it is well written , methodology is well explained in order to be reproducible and bibliography is up to date . In my opinion, it is a good study which merits publication as it gives precious information , strengthen, and advance our knowledge in the field. It should be of high interest to the scientific community , specifically for those working in the area.

Author Response

This is a research article that focuses on the study of confined environments and their impact on human health, especially gut microbiota and its metabolites by advanced combined multi-omic technics. The abundance of Bifidobacterium, Collinsella, Faecalibacterium, and Bacteroides and identification of specific enterotypes in specific confined environments shows how confined environments impact the microbiota composition. Yet, metabolomic analysis showes increase in aminoacids and bile acids pathways. The authors seek to find potential biomarkers for developing targeted strategies to mitigate negative disease effects and protect people working in such environments.

It is a well written paper with all recent bibliography in the field. The experimental part is well designed. Their data were well presented in tables and images and extensively discussed. The researcher's team seems to be qualified and experienced in this scientific area.

The paper could be published in its present as it is well written, methodology is well explained in order to be reproducible and bibliography is up to date. In my opinion, it is a good study which merits publication as it gives precious information, strengthen, and advance our knowledge in the field. It should be of high interest to the scientific community, specifically for those working in the area.

A: We are grateful for your effort reviewing our paper and your positive feedback.

Round 2

Reviewer 1 Report

Comments and Suggestions for Authors

The authors have fulfilled all the requested recommendations, making the manuscript eligible for acceptance.

Author Response

The authors have fulfilled all the requested recommendations, making the manuscript eligible for acceptance.

  • Special thanks to you for your good comments.

Reviewer 2 Report

Comments and Suggestions for Authors

The authors performed the modifications, but They must explain the rationale behind the selection of the cohort under study. The cohort seems underestimated.

Line 89. "The subjects of this study were 45 healthy males, approximately 19-26 years old"

Decrease percent match, please. It should be under 30%.

Comments on the Quality of English Language

Minor editing

Author Response

1.The authors performed the modifications, but They must explain the rationale behind the selection of the cohort under study. The cohort seems underestimated.

A: Thank you for your valuable feedback. The cohort for this study was selected based on the need to control for certain variables that could influence the results, such as age, weight, height, and health status. By selecting healthy male participants within a relatively narrow age range (19-26 years), we aimed to reduce potential confounding factors related to age-related physiological differences. Furthermore, excluding individuals with cardiovascular diseases, skin conditions, or other illnesses ensured that any observed effects were not influenced by pre-existing health issues, which could skew the data. Additionally, the decision to include participants who refrained from alcohol, antibiotics, and other drugs in the days leading up to the experiment was made to prevent these substances from affecting the study outcomes. While the cohort may seem limited, this carefully controlled selection allows for a more precise investigation of the variables under study, reducing noise and enhancing the reliability of the results.

2.Line 89. "The subjects of this study were 45 healthy males, approximately 19-26 years old"

A: Thank you for your comment. Yes, the subjects of this study were 45 healthy males, aged approximately 19-26 years old. This age range was chosen to minimize physiological variability that could occur with age and to focus on a relatively homogenous group for more reliable results. By limiting the study to young, healthy males, we aimed to ensure that the data collected would be less affected by factors such as age-related changes in physiology, ensuring a clearer interpretation of the results related to our research objectives.

3.Decrease percent match, please. It should be under 30%.

A: Thank you for kindly reminding us. We have revised the manuscript.